



# Spatiotemporal response of the water cycle to land use conversions in a typical hilly-gully basin on the Loess Plateau, China

Linjing Qiu[1], Yiping Wu[1], Lijing Wang[1], Xiaohui Lei[2], Weihong Liao[2], Ying Hui[3], Xianyong Meng[4]

[1]Department of Earth and Environmental Science, School of Human Settlements and Civil Engineering, Xi'an Jiaotong University, Xi'an, Shaanxi 710049, China

[2]State Key Laboratory of Simulation and Regulation of Water Cycle in River Basin, China

[3]Meteorological Institute of Shaanxi Province, Xi'an, China

[4]Institute of Water Resources and Hydropower Research, Beijing, 100038, China

*Correspondence to*: Yiping Wu (yipingwu@xjtu.edu.cn)

**Abstract.** The hydrological effects of the 'Grain for Green' project (GFGP) on the Loess Plateau have been largely debated due to the complexity of the water system and its multiple driving factors. The aim of this study was to investigate the response of the hydrological cycle to the GFGP measures based on a case study of the Yanhe basin, a typical hilly-gully area on the Loess Plateau of China. First, we analyzed the land use and land cover (LULC) changes from 1990 to 2010. Then, we evaluated the effects of LULC changes and sloping land conversion on the main hydrological components in the basin considering the land surface characteristics and climate impacts. The Soil and Water Assessment Tool (SWAT) was used for this analysis. The results indicated that farmland exhibited a decreasing trend declining from 40.2% of the basin area in 1990 to 17.6% in 2010, and the woodland and grassland areas correspondingly increased due to the implementation of the GFGP in the basin. Due to land use changes from 1990 to 2010, surface runoff and the water yield exhibited decreasing trends, whereas evapotranspiration (ET) increased, resulting in a persistent decrease in soil water. Additionally, converting cropland areas with slopes ≥15° or >25° to grassland and woodland had negative effects on surface runoff, the water yield and soil water and a positive effect on ET. The magnitudes of the hydrological effects generated by sloping cropland to woodland conversion were greater than those for sloping cropland to grassland conversion. These results suggest that the expansive revegetation of sloping land could reduce runoff generation, particularly in woodland areas, but these effects could reduce the soil water volume in the region. Overall, this study can be used to improve sustainable land use planning and water resource management on the Loess Plateau in China.

## 1 Introduction

Land surface change is one of the most important drivers of eco-hydrological changes (Li et al., 2009; Bloeschl et al., 2007). The impacts of land use and land cover (LULC) changes on water resources and hydrological processes in a river basin are mainly reflected in the overland surface runoff, streamflow, evapotranspiration (ET), soil moisture, etc. (Bari and Smettem, 2004; Chawla and Mujumdar, 2015; Liu et al., 2008b; Zucco et al., 2014). Many studies have investigated the interactive



mechanisms between land use patterns and basin hydrology, and they found that the characteristics of basin hydrology vary among different land use patterns, due not only to different land use types (Wang et al., 2012; Jian et al., 2015; Duan et al., 2016), but also to the spatial heterogeneity of LULC (Chu et al., 2010; Liu et al., 2013). However, debate exists among ecohydrologists regarding the effects of past and ongoing land use changes because of the spatial and temporal complexity of

hydrological processes (Lørup et al., 1998; Lopez-Moreno et al., 2011; Alkama et al., 2013; Liu et al., 2016).

Land use planning in China is considered a crucial strategy for the sustainable management of a river basin system and has been widely adopted for ecological restoration and water resource protection, especially in the Loess Plateau region, which is famous for its fragile ecology (Liu et al., 2008a; Zhang et al., 2009; Zhen et al., 2014). The river basins on the Loess Plateau are important because of the dense population, intensive cultivation, and the high demand for water in the area. Unfortunately,

this region is characterized by insufficient water resources and severe soil erosion, and it has historically experienced vegetation degradation and desertification (Zhao et al., 2013; Guo et al., 2002). Thus, land use change and basin hydrology have attracted a considerable deal of attention, and soil and water conservation practices have been implemented in the area to overcome the ever-increasing environmental problems since the 1970s. In 1999, the 'Grain for Green' project (GFGP) was launched by the Chinese government in the Loess Plateau region, and the primary goal was to retire and convert steep croplands

(slope ≥ 15°) to green lands (Zhou et al., 2012; Liu et al., 2008a). It was reported that the vegetation coverage on the Loess Plateau  increased from 6.5% in the 1970s to 51% in 2010 (Wang et al., 2012), and approximately 16,000 km$^2$ of rain-fed cropland was converted to planted vegetation during the past decade (Feng et al., 2016). Consequently, the hydrological processes on the sloping land and in the river systems have changed, but the extent of these changes and the ties to LULC change remain topics of scientific research.

In the past decades, the hydrological effects of land use change have been widely explored across different spatiotemporal scales on the Loess Plateau. Huang et al. (2003) investigated the runoff response to afforestation using a paired watershed analysis and revealed that forest revegetation reduced annual runoff and the reduction increased with the age of trees. Wei et al. (2015) observed a strong inverse relationship between runoff and increased LULC using statistical analysis from 1997 through 2000. Feng et al. (2016) reported that revegetation increased ET and resulted in a significant (P<0.001) decrease in

the ratio of streamflow to precipitation on the Chinese Loess Plateau. Zuo et al. (2016) combined statistical tests and hydrological modeling to assess the effects of land use  on runoff and found that the water resources in the upstream region decreased more than those in the downstream region. Liang et al. (2015) used an elasticity and decomposition model based on the Budyko framework to simulate and forecast the hydrological effects of ecological restoration and demonstrated that ecological restoration played a dominant role in the reduction of streamflow in 14 main subbasins on the Chinese Loess Plateau.

Another study showed that the GFGP can potentially increase the soil water content and water yield and decrease the runoff and ET (Tian et al., 2016). Moreover, some researchers stated that woody species consume more water by evapotranspiration than do other vegetation types (Wang et al., 2012; Yang et al., 2014), and some studies have documented that large-scale





reforestation has greatly  decreased the water yield and exacerbated water scarcity (Sun et al., 2006), gradually leading to soil

desiccation (Chen et al., 2008; Chen et al., 2007; Wang et al., 2008). In general, these studies indicated that the GFGP on the

Loess Plateau has had an evident influence on basin hydrology; however, because these studies were performed from different

perspectives or concentrated on single hydrological elements, the effects of ecosystem restoration on the water balance have

not been clarified. In addition, most of the studies have been based on statistical methods with short time scales. Although

numerical models are useful tools for quantitatively assessing the hydrological responses to environmental changes, the

existing modeling studies mainly focused on the water discharge in a river channel, and few have analyzed the spatial features

of hydrological responses. Spatially studying such responses could improve watershed management and the development of

strategies for water resource optimization. Therefore, a comprehensive understanding of how LULC change and its spatial

heterogeneity affect the water balance is essential for long-term land use planning and water resource management.

In this study, the Soil and Water Assessment tool (SWAT) was applied to investigate the hydrological impacts of land use

changes, including the sloping land conversion (SLC) program on the Loess Plateau. The Yanhe basin, a main tributary of the

middle reach of the Yellow River that has undergone large-scale revegetation and SLC, was selected to analyze the hydrological

response mechanism using the modeling technique. The specific objectives were as follows: 1) to investigate the

spatiotemporal variations in key water balance components as a result of LULC change, 2) to evaluate the potential effects of

land conversion on water availability under the SLC program, and 3) to examine the changes in soil water storage under

different land use condition. The results provide a useful reference for sustainable land use planning and water resources

management on the Loess Plateau in China.

## 2 Materials and methods

### 2.1 Study area

The Yanhe basin, which is located in northern Shaanxi Province, China (36°21′–37°19′ N and 108°38′–110°29′ E), is a typical

hilly loess area on the Loess Plateau. The drainage area of the basin is 7591 km$^2$, and its elevation rangs from 560 m to 1760

m (Fig. 1). The slope of the basin varies from 0° to 85.3°, with a mean value of 17.7°. The main channel of the Yanhe River is

284.3 km long and originates in Jingbian County. It flows from northwest to southeast through Zhidan County, Ansai County,

Yanan City, and Yanchang County before entering the Yellow River. The soil in the basin developed from loess deposits, and

the dominant soil type is loessial soil which is classified as Calcaric cambisols (FAO 2014). The Yanhe basin characterized by

a semi-arid continental climate with warm and concentrated precipitation in summer and cold, dry winters with occasional

snowfall. The precipitation from 1952–2015 ranged from 300 mm to 803 mm, with a mean annual value of 495 mm.

Additionally, the mean annual maximum and the minimum air temperatures were 17.4 ℃ and 4.2 ℃, respectively. Grassland,

farmland, and woodland (mainly artificial woods) are the dominant land use types in this region. Most crops are cultivated on




sloping lands, and woodlands are generally located on the steeper parts of the landform. The mean annual streamflow at the most downstream station of Ganguyi was $205 \times 10^6$ m$^3$ from 1952 through 2008, and the streamflow from June to September accounted for 64.1% of the total annual discharge at this station. The GFGP was implemented in the basin in 1999, and the observed streamflow exhibited a decreasing trend during the 2010s (Fig. 2).

**2.2 Data**

The SWAT model setup requires daily meteorological data, which were collected from the Meteorological Institute of Shaanxi Province, China. The datasets consist of daily precipitation, maximum and minimum temperature, relative humidity, wind speed and solar radiation, for five county level meteorological stations including Jingbian, Zhidan, Aansai, Yan'an and Yanchang stations. Observed monthly streamflow data at the Ganguyi hydrological station were obtained from the Yellow

River Conservancy Commission (YRCC) from 1980-2010. The data were used for model calibration and validation, although some data gaps (missing values) were present. Ganguyi station is the most downstream station on the Yanhe River and controls an area of 5891 km$^2$. A digital elevation model (DEM) of the basin with a 30-m resolution was obtained from the National Geomatics Center of China (http://www.ngcc.cn/). The soil data, including a raster map with soil property database at a 20-m resolution, were obtained from the Institute of Soil and Water Conservation (ISWC), Chinese Academy of Sciences (CAS).

LULC data from 1990, 2000 and 2010 with a 30-m resolution were supplied by the Institute of Remote Sensing and Digital Earth, CAS. The land area was divided into six main land use types, including grassland, woodland, cropland, water, residential land and barren land, according to the ecosystem classification system of China.

**2.3 Model description**

The SWAT model (version 2012), which was developed by the U.S. Department of Agriculture (USDA)—Agricultural

Research Services (ARS), is a physically based, temporally continuous, and distributed watershed-scale hydrological model (Neitsch et al., 2011; Douglas-Mankin et al., 2010). The SWAT-modeled hydrological cycle is based on the water balance, as documented by Arnold et al. (1998). SWAT has been widely tested and successfully used to explore the effects of climate and land use/management changes on watershed hydrology and water quality (Nyeko, 2015; Panagopoulos et al., 2011; Gassman et al., 2014; Douglas-Mankin et al., 2010; Zhang et al., 2013; Bosch et al., 2010; Wu and Chen, 2013; Xu et al., 2012; Zhang

et al., 2011; Zhang et al., 2008; Qiu et al., 2012). Detailed descriptions of the mechanisms and structure of the SWAT model can be found in several literatures (Neitsch et al., 2011; Arnold et al., 2012).

**2.4 Model setup and calibration/validation**

A geographic information system (GIS) interface, ArcSWAT (Version 2012.10_2.18), was used to set up the model. A 30-m DEM was used to delineate the basin. With the threshold area of subbasins set to 50 km$^2$, this process produced 88 subbasins.

The land use map of 1990 and soil map were used to parameterize the SWAT model. To accurately reflect the spatial variability in the basin, multiple hydrological response units (HRUs) were selected. A single HRU represents a unique combination of





land cover, soil type and slope, and 1136 HRUs were established in the basin. We used the SWAT-CUP module to perform a sensitivity analysis of model parameters, and the SUFI-2 algorithm was used for optimization. SWAT was calibrated using the first ten-year (1983–1992) record of monthly streamflow and validated using data from the subsequent eight years (1993–2000). Additionally, a three-year warm-up period (1980-1982) was used to minimize the effects of uncertain initial conditions

(e.g., soil water storage) in the model simulation. To evaluate the model performance numerically, we used six statistical measures, including percent bias (PBIAS), Nash-Sutcliffe efficiency (NSE) (Nash and Sutcliffe 1970), and the coefficient of determination (squared correlation coefficient, $r^2$). Typically, a model simulation is considered satisfactory with NSE>0.5, -25%≤PBIAS≤25%, and $r^2$>0.5 (Neupane and Kumar, 2015; Wu and Chen, 2013).

**2.5 Modeling scenarios**

In this study, we used three land use conditions (land use maps in 1990, 2000 and 2010, corresponding to LU1990, LU2000 and LU2010, respectively) to evaluate the hydrological impacts of past land use changes from 1986 through 2015. To assess the long-term effects of SLC on the hydrological cycle in the Yanhe basin, the land use condition of 2010 was set as the baseline scenario (BS), and four land conversion scenarios were reestablished based on the land use of 2010 and the SLC policy of the GFGP. During the baseline period, the model was driven by the 30 years (1986-2015) of climate data. Then, we established

four hypothetical SLC scenarios. Scenario 1 (S1) and scenario 2 (S2) refer to the conversion of farmland on slopes steeper than 25° to grassland and woodland, respectively, and scenario 3 (S3) and scenario 4 (S4) assumes that farmland on slopes greater than or equal to 15° is converted to grassland and woodland, respectively. The above four scenarios were implemented using the same climate forcing data to isolate the effects of SLC change.

**3 Results**

**3.1 Model performance evaluation**

We performed a global sensitivity analysis to examine 18 parameters that are potentially related to streamflow and identified the most sensitive 8 parameters for subsequent model calibration (Table 1). The selected 8 parameters represent key hydrological components such as surface runoff, soil water capacity and conductivity, ET, and groundwater recharge in this region. Although the calibrated model slightly underestimated streamflow, the model performance was satisfactory in the

calibration period (1983-1992), with NSE, $r^2$ and PBIAS values of 0.51, 0.71 and 15.7%, respectively (Fig. 3a). In the 8-year (1993-2000) validation period, the model exhibited good performance, with an NSE value of 0.82, although PBIAS still indicated a 16.9% underestimation (Fig. 3b). In both the calibration and validation periods, the underestimation mainly occurred in spring and winter, when little discharge occurs in the watershed. Thus, the model performed poorly during the dry period. Furthermore, a discrepancy can be observed between measured precipitation and streamflow during the dry period

because some small peaks in observed streamflow could not have been caused by the observed precipitation amount. The



potential reasons for this issue include missing precipitation events due to a limited number of stations or errors in streamflow measurements. Nonetheless, both visual comparison and numerical evaluation indicated that the overall model performance was acceptable for simulating the hydrological processes in the Yanhe basin.

**3.2 Land use changes in Yanhe basin from 1990 to 2010**

Farmland, woodland and grassland were the primary land use types in the Yanhe basin, and rapid land use change occurred from 1990 to 2010 (Table 2). Farmland exhibited a decreasing trend, declining from 40.2% of the basin area in 1990 to 17.6% in 2010. Both grassland and woodland exhibited increasing trends, although these trends differed. Grassland increased slowly from 44.7% to 45.3% between 1990 and 2000, followed by a rapid increase from 45.3% to 55.1% between 2000 and 2010. Woodland exhibited an increase of 73.9% from 1990 to 2000 followed by a smaller increase of 7.8% from 2000 to 2010. To

identify the details of the land use conversion patterns, we explored the mutual transition between three major land use types— farmland, grassland, and woodland (Table 3). The transition of farmland to other land use types was observed in the periods of 1990-2000 and 2000-2010, while the conversions of grassland and woodland to other land use types were only evident from 1990-2000. The farmland area decreased dramatically from 1990 to 2010 because it was largely converted to woodland and grassland, with conversion percentages of 16.7% and 39.5% from 1990 to 2000 and 6.4% and 33.2% from 2000 to 2010,

respectively. Spatially, Fig. 4 shows that the conversions mostly occurred in the central and northwestern parts of the basin from 1990–2000 and were scattered from 2000 to 2010. We also examined the land use conversions over the 20-year periods (1990 to 2010) and found that 19.6% and 54.5% of farmland was transformed to woodland and grassland, respectively, indicating evident LULC change due to the implementation of the GFGP policy in the basin (Table 3). Although the percentages of grassland and woodland increased, 20.1% of grassland and 19.5% of woodland were converted into farmland from 1990 to

2000. Additionally, mutual conversions were observed between grassland and other land use types, but the total area of grassland changed only slightly from 1990 to 2000. For example, 1204.6 km$^2$ of farmland and 281.5 km$^2$ woodland were converted into grassland, whereas 762.3 km$^2$ and 683.2 km$^2$ of grassland were transformed to woodlands and farmland, respectively, in same period, resulting in a small change in the net area (Table 3). Additionally, despite the increase in woodland from 1990 to 2010, 12.6% and 31.1% of the woodland from 1990 were converted into farmland and grassland, respectively,

particular in the southern part of the basin.

**3.3 Water balance components under different land use types**

To further analyze the quantitative impacts of LULC changes on the hydrological cycle, the main hydrological components of individual land use types were assessed via model simulation. The analysis results indicated that the water balance varied among different land use types in the basin (Fig. 5). The overland surface runoff per unit area from residential land was the

highest, with an average value of 58.9 mm, followed by those of cropland and grassland, at 19.2 mm and 9.3 mm, respectively. Woodland exhibited the lowest surface runoff of 3.7 mm, which was approximately about 80.7% lower than that of cropland



and 62.2% lower than that of grassland (Fig. 5a). The water yield from individual land use types exhibited a trend similar to that of surface runoff. These results implied that the conversions of cropland and grassland to woodland could reduce runoff and streamflow at the regional scale. By contrast, the ET of woodland areas was the highest, followed by those of grassland, cropland, and residential land. With the exception of residential land, the soil water from other land use types exhibited an

inverse trend compared with that of ET (cropland > grassland > woodland); thus, woodland areas used more soil water for ET and generated less runoff and streamflow. Fig. 5b shows the total water volumes on the different land use types in the basin. The highest volumes of surface runoff and water yield were associated with cropland because of its large water volume per unit area and large total area in the basin. Grassland had the largest soil water storage and highest ET in the basin. The surface runoff, water yield, soil water, and ET of woodland were the lowest among different land use types, with the exception of

residential land, due to the small total area of woodland in 1990.

**3.4 Hydrological impacts of historical land use changes**

The impacts of land use changes were simulated through applying the model to three land use scenarios (LU1990, LU2000, and LU2010) with the same climate forcing data (from 1986 to 2015). The results showed that streamflow at the outlet of the Yanhe basin exhibited negative responses to land use changes between 1990 and 2010, and the magnitude of the decrease

under LU2010 was larger than that under LU2000 (Fig. 6a). Further analysis showed that relatively large responses mainly occurred from May to September (Fig. 6b). Table 4 presents the simulated average annual components of the water balance in the Yanhe basin. Surface runoff generated at the basin scale exhibited a decreasing trend as land use changes occurred from 1990 to 2010, and it decreased by 16.6% in LU2000 and 29% in LU2010. Land use changes had minor effects on subsurface flow, with a slight increase of 1.8% in LU2000 and 2.7% in LU2010 compared with that in LU1990. Similar to the subsurface

flow, ET displayed a weak increasing trend in the basin due to land use changes. The water yield in the basin under LU2000 and LU2010 changed in a similar manner as surface runoff, with decreases of 5.6% and 9.9%, respectively, compared with that of LU1990. The soil water decreased by 11% from 1990 to 2010, with a rapid decrease from 1990-2000 and a slow decrease afterward.

The spatial distributions of the main hydrological components at the HRU scale are shown in Fig. 7. The central part of the

basin exhibited high surface runoff, while most western portion and southern edge of the basin exhibited little runoff (Fig. 7a and 7e). Additionally, the decreasing trend surface runoff mainly occurred on the middle part of the basin from 1990 to 2010. To further understand the hydrological changes associated with land use change, we analyzed the long-term (30-year) average total water volumes of the main hydrological components (Fig. 8). The results showed that decreases in total surface runoff mainly occurred in cropland areas because of the decreased area of cropland. The associated decreased percentages were 26.8%

under LU2000 and 57.3% under LU2010 compared to that of LU1990 (Fig. 8a). Although the total surface runoff generated in grassland and woodland areas increased from LU1990 to LU2010, the total magnitude of these increases could not





compensate for the decreasing associated with cropland. Thus, the total surface runoff of the basin decreased. The water yield exhibited a trend similar to that of surface runoff from LU1990 to LU2010 (Fig. 7b and f)—the total water yield decreased by 60.9% in cropland areas and increased by 18.8% and 136.4% on grassland and woodland, respectively. These changes were due to the dramatic conversion of cropland to grassland and woodland between LU1990 and LU2010 (Fig. 8b). Similar to the

spatiotemporal trends in the water yield, ET increased under LU2010 compared with that of LU1990 (Fig. 7c and g), whereas it decreased by 58.7% in cropland and increased by 21.4% and 124.0% for grassland and woodland from LU1990 to LU2010, respectively (Fig. 8c). In terms of the spatial distribution of soil water, an evident decrease occurred under LU2010, and the highest decrease occurred in the north-central part of the region (Fig. 7d and h). Although the total soil water in woodland exhibited an increasing trend, it was the lowest when compared with the soil water in cropland and grassland areas under the

three land use conditions (Fig. 8d).

### 3.5 Potential impacts of hypothetical SLC

The potential hydrological effects of SLC were projected through inputting four land conversion scenarios that were reconstructed based on the LU2010 baseline (Table 5). The changes in land use in S1 led to the conversion of 131.6 km$^2$ cropland with slope >25° to grassland, which resulted in slight decreases in surface runoff, water yield and soil water, while

subsurface flow and ET exhibited minor increase. With the same acreage of cropland to woodland conversion in S2, the induced hydrological effects were similar to those in S1, whereas the differences between S2 and the baseline were larger in magnitude than those between S1 and the baseline. When the land conversion was extended to cropland with slope ≥15°, the trends of hydrological change were further strengthened. For example, in S3, grassland increased by 23.9% and 15.3% on land surfaces with slopes of 15°-25° and >25 °, resulting in 12.4% and 2.7% decreases in surface runoff and the water yield compared to the

baseline. Conversely, the subsurface flow and ET minimally increased by 2.2% and 0.2%, respectively. Although soil water consistently exhibited a negative response, the decrement was small. When the same area of cropland with slope ≥15° was converted to woodland in S4, the surface runoff and water yield decreased by 17.1% and 6.4% compared to the baseline, while subsurface flow and ET exhibited 3.9% and 0.3% increases, respectively. Soil water notably changed in S4, decreasing by 6.4% compared with the baseline soil water. Fig. 9 illustrates the spatial response of soil water to land use conversion scenarios

compared with the baseline of 2010. Soil water decreases in S4 mainly occurred along the southern edge of the basin and north-central part of the basin, which have a high elevation and slope gradient (Fig. 9d).

### 4 Discussion

Farmland, grassland and woodland were the primary land use classes in the Yanhe basin, and the sum of their area accounted for more than 95% of the total area during our study period. To investigate the effects of land use change and management

practices on the water balance, three land use scenarios were assessed using the SWAT model. The contributions of individual



land use types to basin hydrology differed. Compared to cropland, grassland and woodland areas were generally associated with reduced surface runoff and decreases in the water yield of the basin, particularly woodland. This result is consistent with those of previous studies, which showed that woodland areas captures more rainfall and uptake more water than do other land use types (e.g., cropland and grassland) (Jian et al., 2015; Wang et al., 2012), resulting in reduced infiltration, runoff, and

water discharge (Wang et al., 2013; Duan et al., 2016). Additionally, woodland areas lost more water through ET compared to other vegetation types. This trend was demonstrated by the change in soil water, exhibiting less soil water on the woodland at same precipitation compared to the trends in other areas. In addition, another possible explanation for the low soil water volumes in woodland areas was that forests in the Yanhe basin generally grew on landform with high slopes. Our analysis indicated that more than 62% of woodland was located on slopes ≥15° (Table 5). Steeper slopes generally retain less soil water

due to low infiltration rates and rapid surface drainage (Famiglietti et al., 1998). Thus, a large amount of precipitation was associated with forest growth and ET rather than discharge out of the basin as surface runoff and streamflow. Such water patterns prevent water loss but at the expense of reduced soil water in the region.

After the implementation of the GFGP, the area of cropland decreased continuously because it was transformed into grassland and woodland from 1990 to 2010, and the conversions among land use types were the main forms of land use change

in the Yanhe basin. These results are consistent with previous findings regarding trends in land use change on the Loess Plateau (Li et al., 2016). We compared monthly and annual average streamflow under three land use scenarios, and the results showed that the decreased average monthly streamflow in the rainy season is the primary mechanism for the decrease in annual average streamflow in the basin. In addition, we found that the decrease in surface runoff was the main reason for the streamflow decrease in the basin, and our quantitative evaluation suggested that surface runoff decreased by 29.1% due to land use change

from 1990 to 2010. A similar conclusion was drawn by Farley et al. (2005), who studied 26 watersheds to assess the hydrological effect of afforestation and found that annual runoff was reduced by 31-44% on average. Notably, the average annual volume of soil water exhibited an evident decrease under the 2010 land use scenario. This trend may be due to two interrelated reasons. First, the area of cropland decreased dramatically in 2010, while cropland exhibited the highest soil water volume per unit area relative to other land use types (Fig. 5a), leading to the large decrease in soil water. Second, a certain

proportion of cropland was converted to grassland and woodland, which had the lowest soil water per unit area; thus, the soil water decreased due to the cropland area reduction, and this decrease could not be offset by increases in total soil water in grassland and woodland areas (Fig. 8). Although the area of woodland peaked in 2010, the total soil water in woodland areas only slightly increased. This trend could have occurred because more water was consumed by woodland than cropland and grassland. Additionally, cropland areas with high slopes were converted to woodland after the GFGP was implemented (Zhou

et al., 2012), whereas areas with steep slopes have lower soil moisture retention potentials than areas with gentle slopes (Pachepsky et al., 2001). Thus, topographical factors also played important roles in the spatial heterogeneity of the water balance (Qiu et al., 2001; Bi et al., 2008).





Four future SLC scenarios were proposed to demonstrate the hydrological effects of land use changes. Increase in grassland and woodland cover due to the land conversion of cropland with slopes ≥15° or >25° had negative effects on surface runoff, the water yield and soil water, whereas subsurface flow and ET increased. We found that the magnitudes of the hydrological effects of the conversion of sloped cropland to woodland were greater than those associated with the conversion of sloped

cropland to grassland. This result suggests that the expansion of woodland could reduce runoff generation and drainage because of overland flow retention and intensification of ET in this region. However, the revegetated sloping land was prone to reduced soil moisture due to its steep slope. Some studies have reported that revegetation can cause soil water shortages in both the near-surface soil and deep soil layers (Farley et al., 2005; Jian et al., 2015), likely resulting in soil desiccation (Wang et al., 2011; Fu et al., 2012). Therefore, watershed management should consider all water balance components (Duan et al., 2016),

and vegetative structure and management measures should be optimized to improve the ecohydrological functions and promote the watershed sustainability.

## 5 Conclusions

In this study, we investigated the land use changes from 1990 to 2010 based on a transition analysis. Cropland, grassland and woodland were the dominant land use types in the Yanhe basin, and land use conversion occurred among these types since

1990 due to the implementation of the GFGP. The decrease in cropland led to increase in grassland and woodland. The impacts of LULC changes on the water balance components were assessed quantitatively using the SWAT model and three periods of land use maps and four hypothetical SLC scenarios based on the GFGP policy on the Loess Plateau. Our analysis showed that cropland was associated with the highest surface runoff and water discharge per unit area, followed by grassland and woodland. These differences can partly explain the relationships between hydrological characteristics and land use change at the basin

scale. Surface runoff and water yield exhibited decreasing trends due to land use changes from 1990 to 2010, while subsurface flow and ET increased. Consequently, soil water decreased between 2000 and 2010. By adopting four cropland land conversion scenarios, we found that the function of reducing surface runoff was more effective when croplands with slopes ≥15° were converted into grassland or woodland compared to converting areas with slopes >25°. Furthermore, planting wood on sloping land had greater hydrological effects than did planting grass. Notably, surface runoff and soil water decreased and ET increased.

Overall, this study provides useful information for land use planning and soil and water conservation on the Loess Plateau, and further studies are required to investigate the optimization of the vegetative structure and the avoidance of undesired hydrological effects.

## Acknowledgment

This work was supported by the Thousand Youth Talents Plan (122990901606), the National Natural Science Foundation of




China (41301223), the China Postdoctoral Science Foundation Grant (2016M592777), and State Key Research and Development Project (2016YFC0402208, 2016YFC0401903).

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





**Table 1 Sensitive parameters for streamflow simulation and calibrated values**

| Parameters | Description | Range | Optimized value/ Percent change |
|---|---|---|---|
| CN2 | Soil conservation service (SCS) runoff curve number | -20%~20% | -4.03%[r] |
| SOL_AWC | Soil available water capacity | -20%~20% | 10.71%[r] |
| SOL_K | Soil saturated hydraulic conductivity (mm/h) | -20%~20% | 7.36%[r] |
| ESCO | Soil evaporation compensation factor | 0~1 | 0.51[v] |
| EPCO | Plant uptake compensation factor | 0~1 | 0.65[v] |
| ALPHA_BF | Baseflow alpha factor (day) | 0~0.5 | 0.36[v] |
| GWQMN | Threshold depth of water in the shallow aquifer required for return flow to occur (mm) | 100~1200 | 906.20[v] |
| SURLAG | Surface runoff lag coefficient | 0.05~6 | 2.93[v] |
| REVAPMN | Threshold depth of water in the shallow aquifer for "revap" to occur (mm) | 100~1200 | 499.40[v] |

Note: the superscripts "r" and "v" in the column of optimized value indicate the percent changes based on initial values and replaced parameter values, respectively.





**Table 2 The land use changes in the Yanhe basin from 1990 to 2010**

|  | Farmland | | Grassland | | Woodland | | Residential land | | Water | | Barren land | |
|---|---|---|---|---|---|---|---|---|---|---|---|---|
|  | Area (km$^2$) | Ratio (%) | Area (km$^2$) | Ratio (%) | Area (km$^2$) | Ratio (%) | Area (km$^2$) | Ratio (%) | Area (km$^2$) | Ratio (%) | Area (km$^2$) | Ratio (%) |
| 1990 | 3051.2 | 40.2 | 3390.4 | 44.7 | 1059.6 | 13.9 | 21.4 | 0.28 | 65.4 | 0.86 | 3.1 | 0.040 |
| 2000 | 2246.4 | 29.6 | 3436.3 | 45.3 | 1842.4 | 24.3 | 43.9 | 0.58 | 17.2 | 0.23 | 0.7 | 0.009 |
| 2010 | 1338.3 | 17.6 | 4178.7 | 55.1 | 1986.3 | 26.2 | 66.4 | 0.87 | 16.6 | 0.22 | 0.5 | 0.006 |





**Table 3 The primary patterns of land use transition in the Yanhe basin from 1990 to 2010**

|  | 1990-2000 | | 2000-2010 | | 1990-2010 | |
|---|---|---|---|---|---|---|
|  | Area (km²) | Ratio (%) | Area (km²) | Ratio (%) | Area (km²) | Ratio (%) |
| Farmland to woodland | 509.2 | 16.7 | 143.6 | 6.4 | 598.5 | 19.6 |
| Farmland to grassland | 1204.6 | 39.5 | 745.4 | 33.2 | 1661.8 | 54.5 |
| Farmland to residential land | 15.9 | 0.5 | 20.7 | 0.9 | 28.5 | 0.9 |
| Farmland to barren land | 0.3 | 0.0 | 0.0 | 0.0 | 0.1 | 0.0 |
| Grassland to woodland | 762.3 | 22.5 | 0.6 | 0.0 | 796.7 | 23.5 |
| Grassland to farmland | 683.1 | 20.1 | 1.5 | 0.0 | 410.4 | 12.1 |
| Grassland to residential land | 7.1 | 0.2 | 0.6 | 0.0 | 10.4 | 0.3 |
| Grassland to barren land | 0.4 | 0.0 | 0.0 | 0.0 | 0.3 | 0.0 |
| Woodland to farmland | 206.5 | 19.5 | 0.0 | 0.0 | 133.8 | 12.6 |
| Woodland to grassland | 281.5 | 26.6 | 0.0 | 0.0 | 330.1 | 31.1 |
| Woodland to residential land | 5.2 | 0.5 | 0.5 | 0.0 | 9.4 | 0.9 |
| Woodland to barren land | 0.0 | 0.0 | 0.0 | 0.0 | 0.0 | 0.0 |
| Barren land to woodland | 0.2 | 7.4 | 0.0 | 0.0 | 0.2 | 7.4 |
| Barren land to grassland | 0.8 | 25.9 | 0.0 | 0 | 1.0 | 31.7 |
| Barren land to farmland | 2. 0 | 65.1 | 0.0 | 0.0 | 1.7 | 54.6 |
| Barren land to residential land | 0.0 | 0.0 | 0.0 | 0.0 | 0.2 | 6.4 |



**Table 4 Simulated average annual values of hydrological components in the Yanhe basin under different land use conditions**

|  | Surface runoff (mm) | Subsurface flow (mm) | ET (mm) | Water yield (mm) | Soil water (mm) |
|---|---|---|---|---|---|
| LU1990 | 15.1 | 22.2 | 461.8 | 37.3 | 123.0 |
| LU2000 | 12.6 | 22.6 | 464.5 | 35.2 | 113.1 |
| LU2010 | 10.7 | 22.8 | 466.5 | 33.6 | 109.5 |



**Table 5 Slope characteristics of each land use type in 2010 and slope land conversion scenarios (unit: km$^2$)**

|  |  | Farmland | Woodland | Grassland | Residential land | Barren land |
|---|---|---|---|---|---|---|
| LU2010 | <15° | 814.3 | 761.2 | 1677.7 | 52.5 | 0.3 |
|  | 15°~25° | 392.7 | 808.4 | 1640.8 | 10.8 | 0.1 |
|  | >25° | 131.6 | 417.3 | 860.9 | 3.2 | 0.0 |
| S1 | <15° | — | — | — | — | — |
|  | 15°~25° | — | — | — | — | — |
|  | >25° | 0 | 417.3 | 992.5 | 3.2 | 0.0 |
| S2 | <15° | — | — | — | — | — |
|  | 15°~25° | — | — | — | — | — |
|  | >25° | 0 | 548.9 | 860.9 | 3.2 | 0.0 |
| S3 | <15° | — | — | — | — | — |
|  | 15°~25° | 0 | 808.4 | 2033.5 | 10.8 | 0.1 |
|  | >25° | 0 | 417.3 | 992.5 | 3.2 | 0.0 |
| S4 | <15° | — | — | — | — | — |
|  | 15°~25° | 0 | 1201.0 | 1640.8 | 10.8 | 0.1 |
|  | >25° | 0 | 548.9 | 860.9 | 3.2 | 0.0 |

Note: '—' represents the same value as that of LU2010.



**Table 6 Simulated average annual values of hydrological components under different scenarios of slope land conversion**

|  | Surface runoff (mm) | Subsurface flow (mm) | ET (mm) | Water yield (mm) | Soil water (mm) |
|---|---|---|---|---|---|
| S1 | 10.4 | 22.9 | 466.8 | 33.3 | 109.0 |
| S2 | 10.3 | 23.0 | 466.9 | 33.2 | 107.7 |
| S3 | 9.37 | 23.3 | 467.6 | 32.7 | 107.5 |
| S4 | 8.87 | 23.7 | 467.8 | 32.6 | 102.5 |





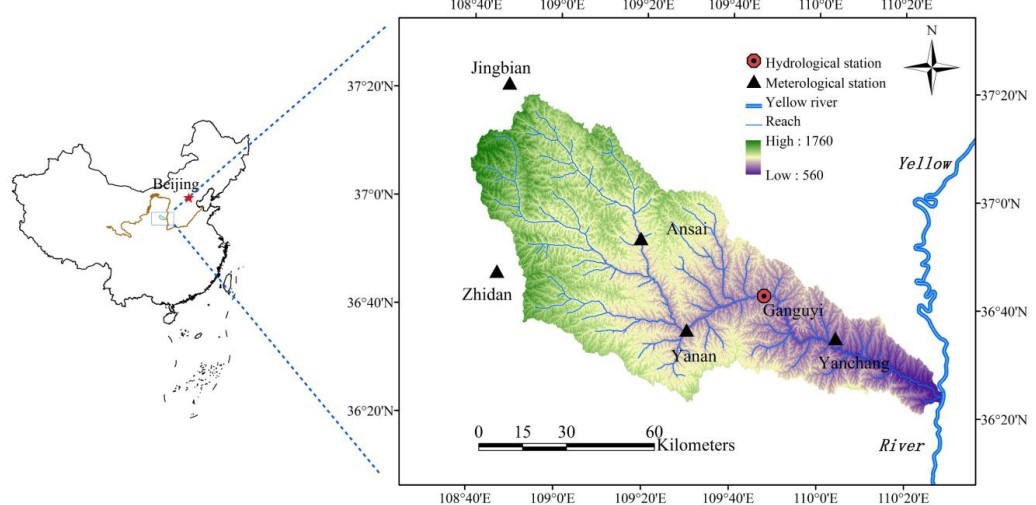

Fig. 1 Map of the location and elevation of the Yanhe basin with the distribution of hydrological station



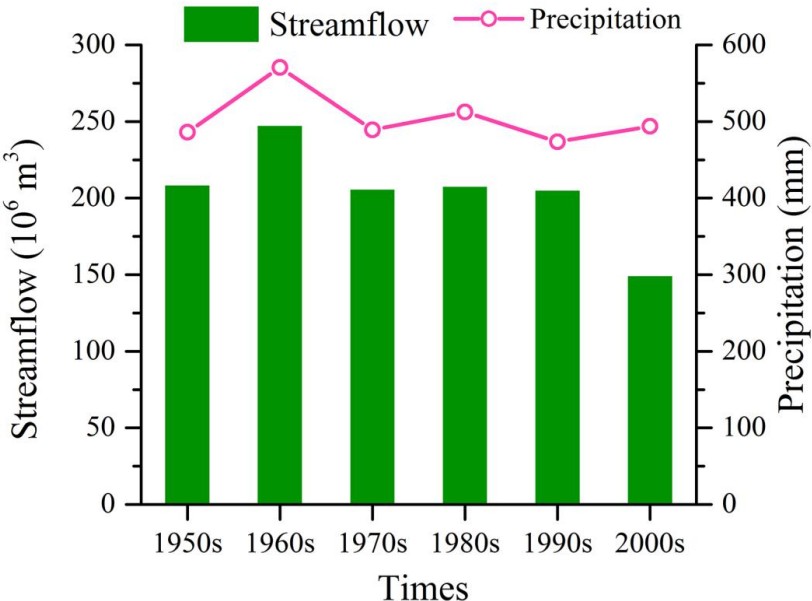

Fig. 2 Precipitation and streamflow changes during different decades in the Yanhe basin




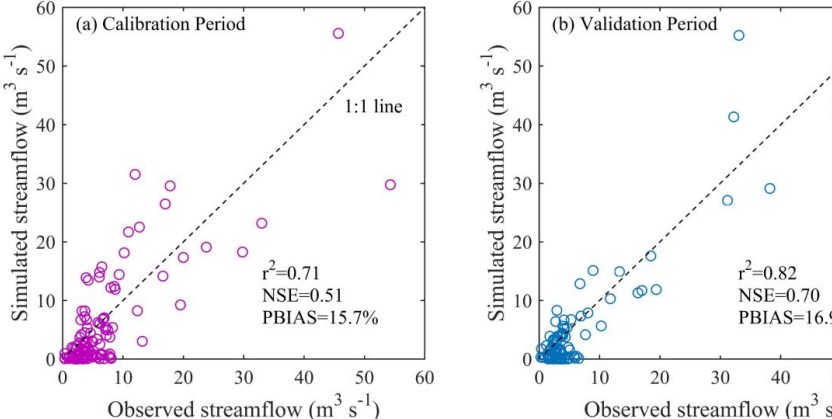

**Fig. 3 The model performance in the (a) 10-year (1983-1992) calibration period and (b) 8-year (1993-2000) validation period**





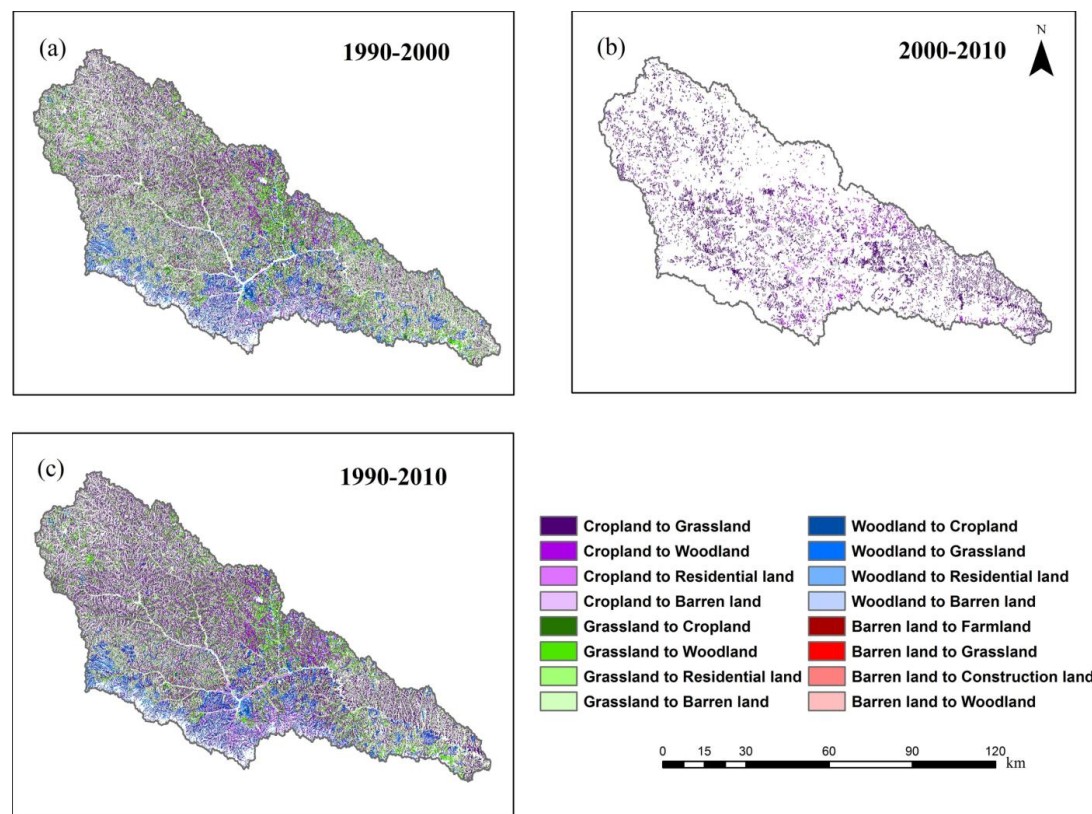

**Fig. 4 Land use transitions from 1990 to 2010**





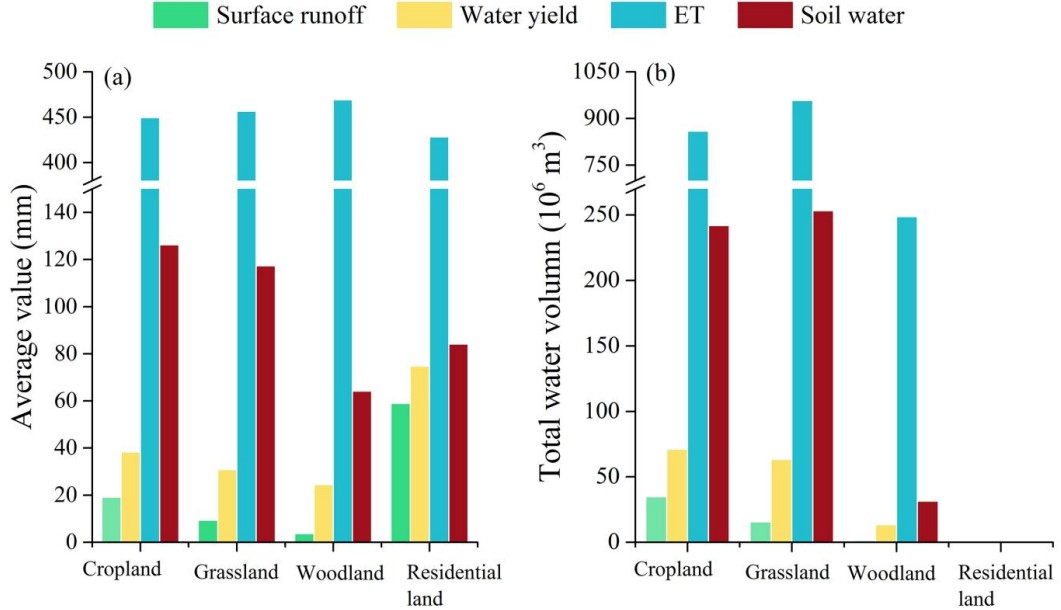

**Fig. 5 Calculated a) unit area and (b) total annual hydrological components in the 18-year simulation period (1983-2000) for different land use types in the Yanhe basin**




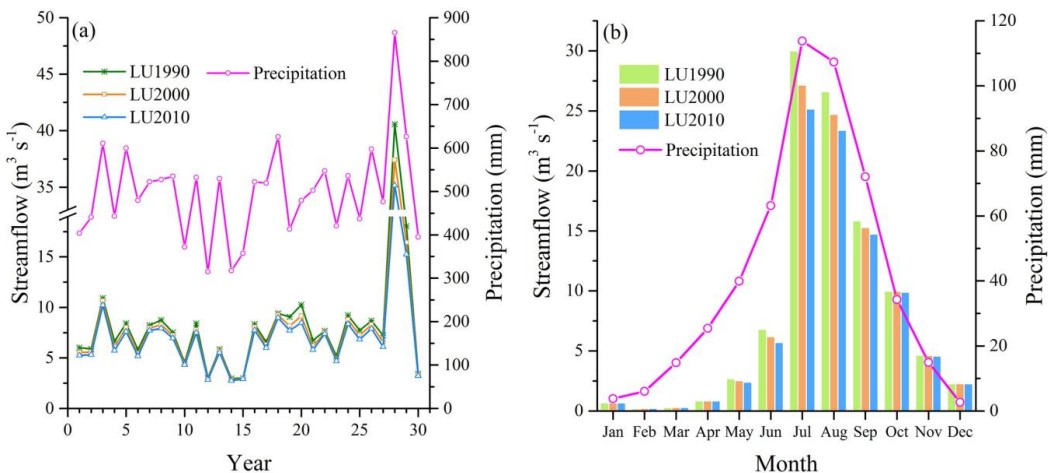

**Fig. 6 (a) Simulated annual streamflow and (b) simulated multiyear average monthly streamflow (the data were averaged from1986 to 2015) at the outlet of the Yanhe basin under different land use conditions**







**Fig. 7 Simulated spatial distribution of (a) surface runoff, (b) water yield, (c) ET and (d) soil water under land use change between 1990 and 2010**





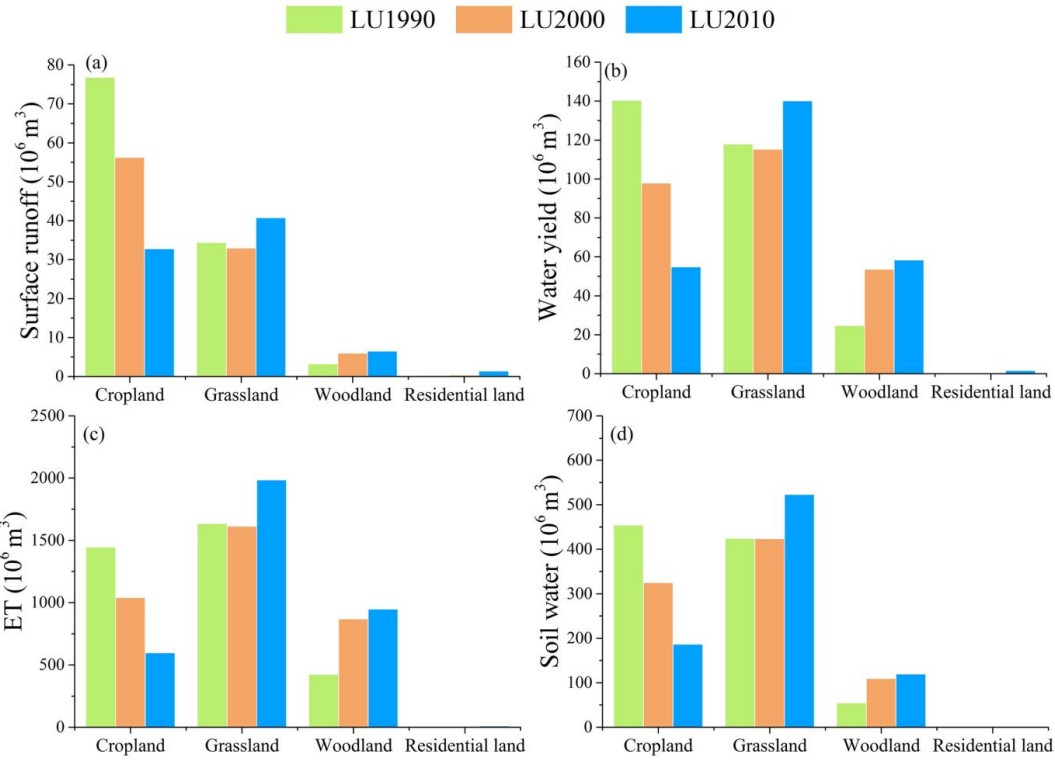

**Fig. 8 Calculated average annual total water volume under different land use scenarios**




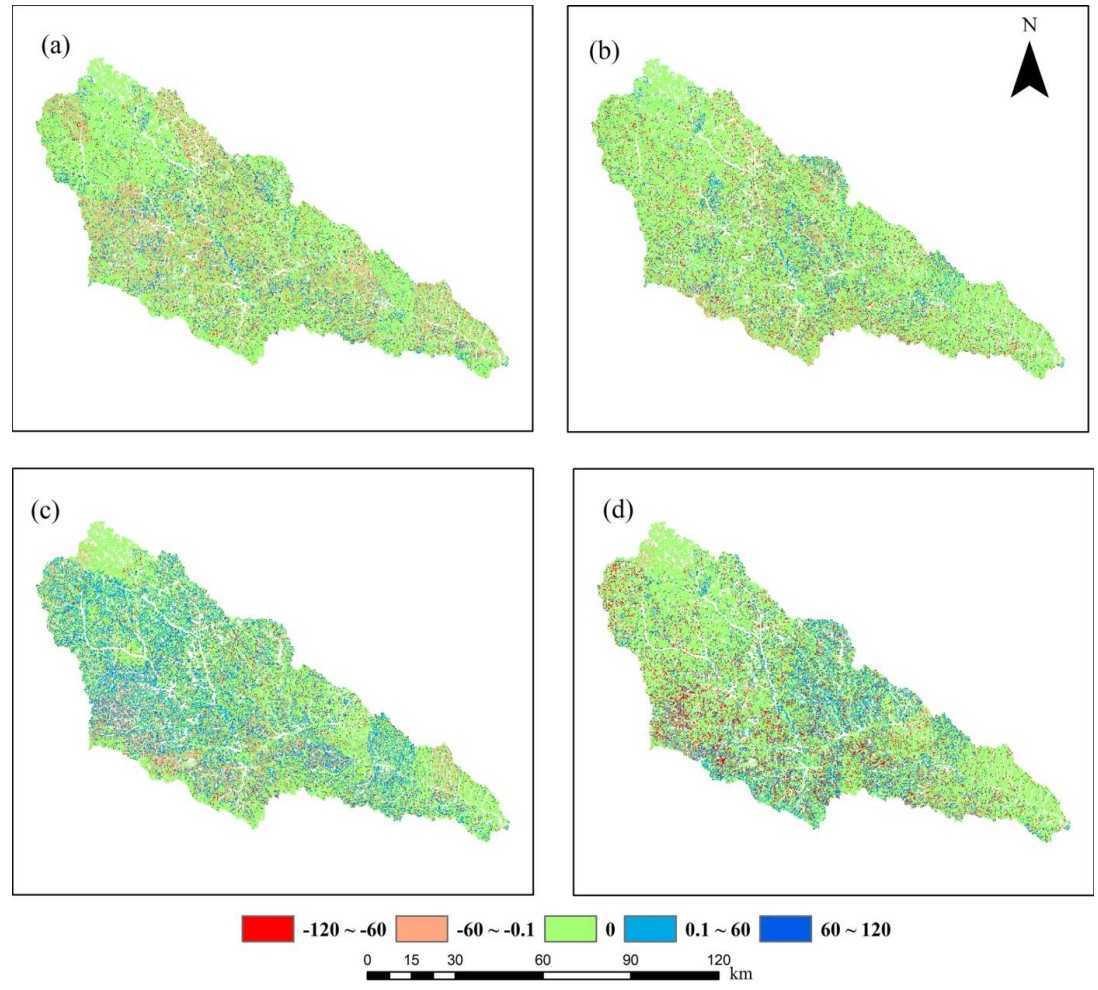

**Fig. 9 Calculated soil water difference between the baseline scenario and a) S1, (b) S2, (c) S3 and (d) S4**