# Peer review of "Spatiotemporal response of the water cycle to land use conversions in a typical hilly-gully basin on the Loess Plateau, China"

_Hydrology and Earth System Sciences, 2017_

## Referee Comment (RC1) · Anonymous Referee #1 · 4 Aug 2017

Qiu et al.- Spatiotemporal response of the water cycle to land use conversions in a typical hilly-gully basin on the Loess Plateau, China- studies the hydrological response of land cover and land use change in a hilly-gully basin of the Loess Plateau in China. This region is very lack of water resources and land use and management play very important role in increasing its resilience and sustainability. This study based on modeling approach adds new and important information for effective water resource conservation and management. Meanwhile it would serve as a useful reference with broad impact for similar arid region worldwide. However, the present form is lack of some details on model construction and validation, especially for the dry period simulation. I would also suggest the author check and improve the input data accuracy. English

word selection and expression accuracy need improvement too. There are too many usage of "exhibit" in the text. The following specific comments provide more suggestion for author to consider in their manuscript revision.

Line 22-23, pp1: "These results suggest that the expansive revegetation of sloping land could reduce runoff generation, particularly in woodland areas, but these effects could reduce the soil water volume in the region." What are the percentages of runoff and soil water volume reductions?

Line 6, pp2: change to "considered as?"

Line 23, pp2: "between runoff and increased LULC" what kind of land cover and land use is that?

Line 27-29, pp3: need valid data sources for the climate information.

Line 4, p4: figure 2 shows that the streamflow is about 80% of precipitation. This is against common sense. Pls check your data again.

Line 10-20, pp5: lulc of 2010 was used as baseline. How about the lulc of 1990 and 2000? Did you use these two maps as model calibration and validation? It seems only 2010 map was used for the slope related scenario creation. It is a little bit confusing here. You may need to clarify more on the model and scenario configurations.

Line 29-30, pp5: do you mean the precipitation input miss the streamflow simulations? If so, you may need to prove/show that in a new figure.

Table3: change "Ratio" to "Percentage" for consistence.

Line 23, pp6: it should be "in the same period" and "resulting the small net change"

Line 28, pp8: change "classes" to "types"

Line 11, pp9: what the difference between surface runoff and streamflow?

Line 23, pp9: why cropland has the highest soil water storage?

[Figure]

**HESSD**

Line 4, pp10: sloped cropland or sloping cropland? Pls make this term consistent throughout the manuscript.

Line 13, pp10: "transition analysis"? Pretty new and never heard. Pls consider changing it. Otherwise more definition should be given.

Line 24, pp10: change planting wood to afforestation?

The study (Qiao et al., 2017, Woody plant encroachment reduce annual runoff and shifts runoff mechanisms in the tallgrass prairie, USA. Water Resource Research) provides some evidence of water budget difference between woodland and grassland and would be a useful reference for this study.

---

## Referee Comment (RC2) · Anonymous Referee #2 · 5 Aug 2017

General Comment: This manuscript investigated the response of main hydrological components to land use change and basin management policy in the loess region of China. It is interesting to see the spatial distribution of hydrological responses at the unit (HRU) level and the quantitative assessment of the roles of different land use types in basin hydrology. In particular, it is attractive to interpret the mechanism of the streamflow decrease associated with vegetation restoration on the Loess Plateau and thus may have a broad readership. Overall, the article was well organized, well written, and easy to follow. All figures and tables are of high quality and informative. What I would suggest is give more discussion on the reason why the different land use types show different hydrological characteristics (e.g., soil water and ET) in this region. That

would help reader better understand the findings from the article.

Specific comments: P2 L6: Use "sloping cropland" here to guarantee the consistence, because you use "sloping land" in the whole manuscript. P7 L26 Add "of" between "trend" and "surface runoff". P7 L29 Replace "because of the decreased area of cropland" by "because of its decreased area". P8 L1 Replace "deceasing" by "decrease". P9 L1-12 the response of soil water in woodland are well discussed, how about in cropland and grassland and try to explain more. P9 L14 Rephrase the sentence and clearly state what are the main forms of land use change in the Yanhe basin. P9 L20 Where are the locations of 26 watersheds? P18: "Ratio" should be changed to "Percentage" in Table 3.

---

## Author Comment (AC1) · 5 Sep 2017

We thank two reviewers very much for their insightful comments and suggestions. All comments are valuable and helpful for improving the quality of the manuscript and have been carefully checked and addressed. Please see our point-to-point responses below. A kindly note: reviewers' comments are in blue color and our responses are in black color; page and line numbers cited in our responses refer to the revised manuscript with track changes for easy evaluation. A track-changed version of the revised manuscript can be found after the response to reviewers.

**Reviewer #1:**

Qiu et al.- Spatiotemporal response of the water cycle to land use conversions in a typical hilly-gully basin on the Loess Plateau, China- studies the hydrological response of land cover and land use change in a hilly-gully basin of the Loess Plateau in China. This region is very lack of water resources and land use and management play very important role in increasing its resilience and sustainability. This study based on modeling approach adds new and important information for effective water resource conservation and management. Meanwhile it would serve as a useful reference with broad impact for similar arid region worldwide. However, the present form is lack of some details on model construction and validation, especially for the dry period simulation. I would also suggest the author check and improve the input data accuracy. English word selection and expression accuracy need improvement too. There are too many usage of "exhibit" in the text. The following specific comments provide more suggestion for author to consider in their manuscript revision.
**Response:** We thank for the positive evaluation and instructive comments and suggestions.

(1) According to the reviewer's suggestion, we add more details on model construction, and this procedure is as follows: a geographic information system (GIS) interface, ArcSWAT (Version 2012.10_2.18), was used to set up the model. The DEM was used to delineate the watershed, and this process produced 88 subbasins. The 30 m land use map of 1990 and soil map were from the Chinese Academy of Sciences, and the digitalized soil data were from the Data Center of West China (http://westdc.westgis.ac.cn/). We used the multiple Hydrological Response Unit (HRU) option in ArcSWAT to represent each unique field (combination of land cover, soil, and slope) as a separate unit, leading to a total of 1,136 HRUs using a threshold value of 10% for the three categories. To meet the needs of the model, the SWAT codes for main land use types in the study area were defined as cropland (AGRR), woodland (FRST), grassland (RNGE), residential land (URBN), barren land (BARR) in the attribute data (*Please see P6 L19 ~ P7 L4*).

(2) In terms of model verification, we firstly calibrated the SWAT model with 10-year (1983–1992) monthly streamflow data of Ganguyi hydrological station using SWAT-CUP (SUFI2 algorithm). Then the model was validated using data from the subsequent eight years (1993–2000). To evaluate the model performance numerically, we used three statistical measures, including percent bias (PBIAS), Nash-Sutcliffe efficiency (NSE), and the coefficient of determination (squared correlation coefficient, $r^2$). Typically, a model simulation is considered satisfactory with NSE>0.5, -25%≤PBIAS≤25%, and $r^2$>0.5 (Wu and Chen, 2013; Neupane and Kumar, 2015) (*Please see P7 L4 ~ P7 L14*). In the calibration and validation periods, the statistics of NSE, PBIAS and $r^2$ ranged from 0.51~0.70, 15.7%~16.9% and 0.71~0.82, respectively, indicating a satisfactory performance. However, PBIAS value showed that SWAT

underestimated the streamflow in both calibration and validation periods. Visual inspection as shown in Fig. 3 indicated that the underestimation mainly occurred in the dry period, and the potential reasons are measurement quality and model behavior for dry areas. Measurement quality refers to the missing precipitation data or streamflow measuring error. The study area belongs to the hilly-gullied loess region, and complex and rugged terrain results in highly variable precipitation and difficulties in spatially estimating the precipitation (Cao et al., 2006). Additionally, this region has limited meteorological and hydrological stations. In this study, there are five meteorological stations, whereas only three stations are located inside the basin and the remaining two are located outside the basin (Fig. 1). It has been reported that the accuracy of the streamflow prediction mainly depends upon the precipitation gage numbers and their locations (Cao et al., 2006; Mul et al., 2009). Thus, the insufficient precipitation records and relatively distant geographical position of meteorological stations resulted in poor streamflow simulation. "Model deficiency" refers to the model limitations such as inadequate representation of physical mechanism of hydrological processes. For example, SWAT uses total daily precipitation and does not consider rainfall intensity within a day, thus it can result in underestimation of streamflow for some precipitation events (Qiu et al., 2012). Another example is the use of runoff curve number to simulate the surface runoff behavior, and this does not account for saturation excess runoff or contributions from variable source areas (Garen and Moore, 2005; Easton et al., 2008). Although the simulation results in our research met the calibration criteria, the model performance can be improved further if precipitation from additional gages available in the basin. Above information was added in the paragraph one of the discussion section (*Please see P12 L9 ~ P13 L3*).

(3) We thank the reviewer for pointing out our language problems (e.g., too many use of a word). Now we have checked the manuscript carefully and revised it further to improve its quality and accuracy.

Line 22-23, pp1: "These results suggest that the expansive revegetation of sloping land could reduce runoff generation, particularly in woodland areas, but these effects could reduce the soil water volume in the region." What are the percentages of runoff and soil water volume reductions?
**Response:** Following the reviewer's suggestion, we calculated the decrease percentage. When the cropland with slope $\geq 15°$ was converted to woodland in S4, the surface runoff and soil water decreased by 17.1% and 6.4%, respectively, compared to the land use condition of 2010. We added this information in the abstract (*please see P1 L22 ~ P1 L23*).

Line 6, pp2: change to "considered as?"
**Response:** Done (*please see P2 L17*).

Line 23, pp2: "between runoff and increased LULC" what kind of land cover and land use is that?
**Response:** "LULC" should be changed to "land cover", and thanks for pointing out this writing error. In the literature, it was reported that mean runoff in 2005–2010 declined significantly compared with the time period of 1997–2005, and this can be attributed mainly to changes in land use/land cover, i.e., increases in forests, shrubs, and grasses, and decreases in sloping farmlands. Thus, we rephrased the sentence to make the description more accurate (*please see P3 L11 ~ P3 L13*).

Line 27-29, pp3: need valid data sources for the climate information.

**Response:** The climate information of study area was based on the collected observation data, and the data was provided by the Meteorological Institute of Shaanxi Province, China. We added the data source in the manuscript accordingly (*please see P5 L1*).

Line 4, p4: figure 2 shows that the streamflow is about 80% of precipitation. This is against common sense. Pls check your data again.

**Response:** The units of the streamflow ($10^6$ m$^3$) and precipitation (mm) are different, and the calculated ratio of streamflow to precipitation is around 6.6%.

Line 10-20, pp5: lulc of 2010 was used as baseline. How about the lulc of 1990 and 2000? Did you use these two maps as model calibration and validation? It seems only 2010 map was used for the slope related scenario creation. It is a little bit confusing here. You may need to clarify more on the model and scenario configurations.

**Response:** We have added more details on model set up and scenario configurations based on the reviewer's comments. We firstly calibrated the SWAT model using land use data of 1990. Then we test different land use data of 1990, 2000 and 2010 under the same climate data from 1986-2015 to separate the unique effect of land use change. Finally, the hydrological effects of hypothetical land use conversion were assessed using same procedures with evaluating historic land use change effects (*please see P6 L18 ~ P7 L24*).

Line 29-30, pp5: do you mean the precipitation input miss the streamflow simulations? If so, you may need to prove/show that in a new figure.

**Response:** Indeed, here we tried to explain the reason for the discrepancy between measured precipitation and streamflow during the dry period. We deleted this sentence and gave the explanations in the section of Discussion (*please see P12 L9 ~ P13 L3*).

Table 3: change "Ratio" to "Percentage" for consistence.

**Response:** Done (*please see P25 Table 3*)**.**

Line 23, pp6: it should be "in the same period" and "resulting the small net change"

**Response:** Done (*please see P9 L13*).

Line 28, pp8: change "classes" to "types"

**Response:** Done (*please see P13 L5*).

Line 11, pp9: what the difference between surface runoff and streamflow?

**Response:** In SWAT, surface runoff is flow that occurs along the sloping surface (i.e., quick-response overland flow), which contributed to streamflow. It occurs whenever the net rainfall exceeds the rate of infiltration or the it was saturated.

Streamflow is the water flowing downhill through creeks, streams, and rivers toward the outlet of the basin, and it includes surface runoff, lateral flow within soil profile, and base flow.

**Line 23, pp9: why cropland has the highest soil water storage?**

**Response:** The reason can be explained as follows: Firstly, in the Yanhe basin, the cropland was mainly distributed in the bottom region of the valley or the sloping land with gentle slope. The valley bottom generally receives more water flow from hillslope and have lower temperature, resulting in more soil moisture (Wang et al., 2012). Additionally, gentle sloping area had higher soil water retention ability compared with steep sloping area (Famiglietti et al., 1998). Secondly, our results showed that the cropland had lower evapotranspiration per unit area compared with other land use types. Thus, these interrelated above two reasons led to the highest soil water storage. This explanation has been added to the revised manuscript (*please see P13 L8 ~ P13 L15*).

**Line 4, pp10: sloped cropland or sloping cropland? Pls make this term consistent throughout the manuscript.**

**Response:** Thanks for the suggestion, and we used "sloping cropland" in the whole manuscript (*please see P15 L3*).

**Line 13, pp10: "transition analysis"? Pretty new and never heard. Pls consider changing it. Otherwise more definition should be given.**

**Response:** "Land us transition analysis" summarizes the amount of certain land that changes from each category at the initial time to each category at the subsequent time, and it has been widely used in landscape ecology and GIS studies of land use to quantitatively estimate the rate of change. To avoid confusion, we took "land use change analysis" in the revision (*please see P8 L24 ~ P9 L1*).

**Line 24, pp10: change planting wood to afforestation?**

**Response:** Done (*please see P15 L23*).

**The study (Qiao et al., 2017, Woody plant encroachment reduce annual runoff and shifts runoff mechanisms in the tallgrass prairie, USA. Water Resource Research) provides some evidence of water budget difference between woodland and grassland and would be a useful reference for this study.**

**Response:** Thanks for recommending a valuable reference to our study, and we have cited this article in our revision (*please see P13 L20*).

**Reviewer #2:**

General Comment: This manuscript investigated the response of main hydrological components to land use change and basin management policy in the loess region of China. It is interesting to see the spatial distribution of hydrological responses at the unit (HRU) level and the quantitative assessment of the roles of different land use types in basin hydrology. In particular, it is attractive to interpret the mechanism of the streamflow decrease associated with vegetation restoration on the Loess Plateau and thus may have a broad readership. Overall, the article was well organized, well written, and easy to follow. All figures and tables are of high quality and informative. What I would suggest is give more discussion on the reason why the different land use types show different hydrological characteristics (e.g., soil water and ET) in this region. That would help reader better understand the findings from the article.

**Response:** We thank the reviewer for his/her positive remarks and constructive comments. We considered the comments of reviewer and added more discussion on hydrological characteristics. For soil water, explanations for differences among different land use types need to combine the topography and land cover on the land surface. In our study, woodland had lowest surface runoff and water yield, because woodland areas capture more rainfall and uptake more water than other land use types (e.g., cropland and grassland) (Wang et al., 2012; Jian et al., 2015), resulting in a lower infiltration, runoff, and water discharge (Wang et al., 2013; Duan et al., 2016). Less soil water on woodland is because forests in the Yanhe basin generally grew on landform with high slopes, and steeper slopes generally retain less soil water due to low infiltration (Famiglietti et al., 1998). Moreover, woodland may lose more water through ET than other land use types. Cropland had highest soil water in our study, and this can be partly attributed to the topography of croplands. The primary croplands are situated in the bottom region of the valley or the sloping land with gentle slope, and valley bottom generally receives more water flow from hillslope and have lower temperature, resulting in more soil water (Wang et al., 2012). Moreover, gentle sloping area had higher soil water retention ability compared with steep sloping area (Panagopoulos et al., 2011). Besides, the cropland had lower evapotranspiration per unit area compared with other land use types. In contrast, grassland had lower soil water, and this can be because grassland had higher soil water loss through evapotranspiration. Wang et al. (2012) reported that grass cover types cannot protect the soil surface from solar radiation, leading to greater water loss via direct evaporation. As for ET on the Loess Plateau region, it has been widely demonstrated that the woodland had highest ET, followed by grassland, and cropland (Xiao et al., 2013; Wang et al., 2012). (*please see P13 L8 ~ P14 L4*)

Specific comments:

P2 L6: Use "sloping cropland" here to guarantee the consistence, because you use "sloping land" in the whole manuscript.
**Response:** Corrected (*please see P15 L3-4*).

P7 L26 Add "of" between "trend" and "surface runoff".
**Response:** Done (*please see P10 L3*).

P7 L29 Replace "because of the decreased area of cropland" by "because of its decreased area".
**Response:** Corrected (*please see P11 L2*).

P8 L1 Replace "deceasing" by "decrease".
**Response:** Corrected (*please see P11 L5*).

P9 L1-12 the response of soil water in woodland are well discussed, how about in cropland and grassland and try to explain more.
**Response:** As responded to the reviewer #1, highest soil water in cropland may be due to two related reasons: Firstly, in the Yanhe basin, the primary croplands were situated in the bottom region of the valley or the sloping land with gentle slope, and valley bottom generally receives more water flow from hillslope and have lower temperature, resulting in more soil moisture (Wang et al., 2012); gentle sloping area had higher soil water retention ability compared with steep sloping area (Panagopoulos et al., 2011). Secondly, our results showed that the cropland had lower evapotranspiration per unit area compared with other land use types. Thus, these interrelated two main reasons led to the highest soil water storage in cropland. In contrast, grassland had lower soil water, and this can be because grassland had higher soil water loss through evapotranspiration. Wang et al. (2012) reported that grass cover types cannot protect the soil surface from solar radiation, leading to greater water loss via direct evaporation (*please see P13 L8-19*).

P9 L14 Rephrase the sentence and clearly state what are the main forms of land use change in the Yanhe basin.
**Response:** It was rephrased, and the revised sentence is "After the implementation of the GFGP, the area of cropland decreased continuously because it was transformed into grassland and woodland from 1990 to 2010, and the conversions among cropland, grassland and woodland were the main forms of land use change in the Yanhe basin" (*please see P14 L5-6*).

P9 L20 Where are the locations of 26 watersheds?
**Response:** This study reported in the literature was conducted globally. The researchers selected 26 catchments around the world and evaluated the effect of afforestation on water yield. To avoid confusion, we added "globally" in this sentence (*please see P14 L12*).

P18: "Ratio" should be changed to "Percentage" in Table 3.
**Response:** Corrected (*please see P25 Table3*).

*Correspondence to*: Yiping Wu (yipingwu@xjtu.edu.cn)

**Abstract.** The hydrological effects of the 'Grain for Green' project (GFGP) on the Loess Plateau have been largely debated due to the complexity of the water system and its multiple driving factors. The aim of this study was to investigate the response of the hydrological cycle to the GFGP measures based on a case study of the Yanhe basin, a typical hilly-gully area on the Loess Plateau of China. First, we analyzed the land use and land cover (LULC) changes from 1990 to 2010. Then, we evaluated the effects of LULC changes and sloping land conversion on the main hydrological components in the basin using Soil and Water Assessment Tool (SWAT) . The results indicated that  cropland exhibited a decreasing trend declining from 40.2% of the basin area in 1990 to 17.6% in 2010, and the woodland and grassland areas correspondingly increased With the land use changes from 1990 to 2010,  the water yield  showed a decreasing trend which was mainly caused by surface runoff decrease, whereas evapotranspiration (ET) had an increasing trend, resulting in a persistent decrease in soil water. Converting sloping cropland  to grassland or woodland exerted negative effects on  water yield and soil water . Particularly when cropland with slope ≥15° was converted to woodland, reduction effects were most evident, and the surface runoff and soil water decreased by 17.1% and 6.4% compared with land use condition of 2010, respectively.

  These results suggest that the expansive  reforestation  on sloping land in the loess hilly-gully region could  decrease the water yield and increase the ET, and consequently caused the soil water reduction . 
[revised manuscript text omitted]

---

## Author Comment (AC2) · 5 Sep 2017

The response to comments was uploaded in the form of a supplement.

Please also note the supplement to this comment:
https://www.hydrol-earth-syst-sci-discuss.net/hess-2017-343/hess-2017-343-AC2-supplement.pdf

———————————————————

---

## Author Response (AR1)

**Dear Editor,**

**Thank you very much for your constructive comments. We took them into consideration when revising the manuscript. Regarding your comments 'the clarifications which are needed to make it easier to understand exactly what you have done', we added more information in the section of method (2.4, 2.5, and 2.6), which will better explain the purpose and research procedures of our work. Additionally, we have revised discussion section and added more explanation to clarify the mechanism of hydrological effect of land use change. As to the English word selection and expression, we got help from a native speaker to check the manuscript, and all writing error have been corrected. We believe that all of these revisions improved the quality substantially. Thanks again for your time.**

**Responses to reviewers**

We thank two reviewers very much for their insightful comments and suggestions. All comments are valuable and helpful for improving the quality of the manuscript and have been carefully checked and addressed. Please see our point-by-point responses below. A kindly note: reviewers' comments are in blue color and our responses are in black color; page and line numbers cited in our responses refer to the revised manuscript with track changes for easy evaluation. A track-changed version of the revised manuscript can be found after the response to reviewers.

**Reviewer #1:**

Qiu et al.- Spatiotemporal response of the water cycle to land use conversions in a typical hilly-gully basin on the Loess Plateau, China- studies the hydrological response of land cover and land use change in a hilly-gully basin of the Loess Plateau in China. This region is very lack of water resources and land use and management play very important role in increasing its resilience and sustainability. This study based on modeling approach adds new and important information for effective water resource conservation and management. Meanwhile it would serve as a useful reference with broad impact for similar arid region worldwide. However, the present form is lack of some details on model construction and validation, especially for the dry period simulation. I would also suggest the author check and improve the input data accuracy. English word selection and expression accuracy need improvement too. There are too many usage of "exhibit" in the text. The following specific comments provide more suggestion for author to consider in their manuscript revision.
**Response:** We thank for the positive evaluation and instructive comments and suggestions.
(1) According to the reviewer's suggestion, we added more details on model construction, and this procedure is as follows:
A geographic information system (GIS) interface, ArcSWAT (Version 2012.10_2.18), was used to set up the model. The DEM was used to delineate the watershed, and this process produced 88 subbasins. The 30 m land use map of 1990 and soil map were from the Chinese Academy of Sciences, and the digitalized soil data were from the Data Center of West China (http://westdc.westgis.ac.cn/). We used the multiple Hydrological Response Unit (HRU) option in ArcSWAT to represent each unique field (combination of land cover, soil, and slope) as a separate unit, leading to a total of 1,136

HRUs using a threshold value of 10% for the three categories. To meet the needs of the model, the SWAT codes for the main land use types in the study area were defined as cropland (AGRR), woodland (FRST), grassland (RNGE), residential land (URBN), and barren land (BARR) in the attribute data (*Please see P6 L22 ~ P7 L7*).

(2) In terms of model verification, we firstly calibrated the SWAT model with 10-year (1983–1992) monthly streamflow data of Ganguyi hydrological station using SWAT-CUP (SUFI2 algorithm). Then the model was validated using data from the subsequent eight years (1993–2000). To evaluate the model performance numerically, we used three statistical measures, including percent bias (PBIAS), Nash-Sutcliffe efficiency (NSE), and the coefficient of determination (squared correlation coefficient, $r^2$). Typically, a model simulation is considered satisfactory with NSE>0.5, -25%≤PBIAS≤25%, and $r^2$>0.5 (Wu and Chen, 2013; Neupane and Kumar, 2015) (*Please see P7 L7 ~ P7 L17*). In the calibration and validation periods, the statistics of NSE, PBIAS and $r^2$ ranged from 0.51~0.70, 15.7%~16.9% and 0.71~0.82, respectively, indicating a satisfactory performance. However, PBIAS value showed that SWAT underestimated the streamflow in both calibration and validation periods. Visual inspection as shown in Fig. 3 indicated that the underestimation mainly occurred in the dry period. The potential reasons for the underestimation can be categorized as "measurement quality" and "model behavior" reasons. "Measurement quality" refers to missing precipitation data or streamflow measuring error. The study area lies in the hilly-gullied loess region, and the complex and rugged terrain results in highly variable precipitation and difficulties in spatially estimating the precipitation (Cao et al., 2006). Additionally, this region has a limited number of meteorological and hydrological stations. In this study, five meteorological stations were involved; however, only three stations are located inside the basin, with the remaining two occurring outside the basin (Fig. 1). It has been reported that the accuracy of streamflow prediction mainly depends upon the precipitation gage numbers and their locations (Cao et al., 2006; Mul et al., 2009). Thus, the insufficient precipitation records and the distances among the meteorological stations resulted in poor streamflow simulation. "Model behavior" refers to model limitations such as the inadequate representation of the physical mechanisms of hydrological processes. For example, SWAT uses total daily precipitation and does not consider rainfall intensity within a day; thus, it can underestimate streamflow for some precipitation events (Qiu et al., 2012). Another example is the use of runoff curve number to simulate the surface runoff behavior. This approach does not account for saturation excess runoff or contributions from variable source areas (Garen and Moore, 2005; Easton et al., 2008). Although the simulation results in our research met the calibration criteria, the model performance can be improved if precipitation data from additional gages become available in the basin. Above information was added in the paragraph one of the discussion section (*Please see P13 L10 ~ P14 L5*).

(3) We thank the reviewer for pointing out our language problems (e.g., too many use of a word). Now we have checked the manuscript carefully and revised it further to improve its quality and accuracy.

Line 22-23, pp1: "These results suggest that the expansive revegetation of sloping land could reduce runoff generation, particularly in woodland areas, but these effects could reduce the soil water volume in the region." What are the percentages of runoff and soil water volume reductions?

**Response:** Following the reviewer's suggestion, we calculated the decrease percentage. Compared with the land use

condition in 2010, the negative effects were most evident where cropland with a slope $\geq 15°$ was converted to woodland, with decreases in surface runoff and soil water of 17.1% and 6.4%, respectively. We added this information in the abstract (*please see P1 L22 ~ P1 L24*).

Line 6, pp2: change to "considered as?"
**Response:** Done (*please see P2 L19*).

Line 23, pp2: "between runoff and increased LULC" what kind of land cover and land use is that?
**Response:** In the literature, it was reported that mean runoff in 2005–2010 declined significantly compared with the time period of 1997–2005, and this can be attributed mainly to changes in land use/land cover, i.e., increases in forests, shrubs, and grasses, and decreases in sloping farmlands. Thus, we rephrased the sentence to make the description more accurate (*please see P3 L13-15*).

Line 27-29, pp3: need valid data sources for the climate information.
**Response:** The climate information of study area was based on the collected observation data, and the data was provided by the Meteorological Institute of Shaanxi Province, China. We added the data source in the manuscript accordingly (*please see P5 L4*).

Line 4, p4: figure 2 shows that the streamflow is about 80% of precipitation. This is against common sense. Pls check your data again.
**Response:** The units of the streamflow ($10^6$ $m^3$) and precipitation (mm) are different, and the calculated ratio of streamflow to precipitation is around 6.6%.

Line 10-20, pp5: lulc of 2010 was used as baseline. How about the lulc of 1990 and 2000? Did you use these two maps as model calibration and validation? It seems only 2010 map was used for the slope related scenario creation. It is a little bit confusing here. You may need to clarify more on the model and scenario configurations.
**Response:** We have added more details on model set up and scenario configurations based on the reviewer's comments. We firstly calibrated the SWAT model using land use data of 1990. Then we test different land use data of 1990, 2000 and 2010 under the same climate data from 1986-2015 to separate the unique effect of land use change. Finally, the hypothetical land use conversion scenarios were developed from land use condition of 2010 and its hydrological effects were assessed using same procedures with evaluating historic land use change effects (*please see P6 L21 ~ P8 L18*).

Line 29-30, pp5: do you mean the precipitation input miss the streamflow simulations? If so, you may need to prove/show that in a new figure.
**Response:** Indeed, here we tried to explain the reason for the discrepancy between measured precipitation and streamflow during the dry period. We deleted this sentence and gave the explanations in the section of Discussion (*please see P13 L11 ~ P14 L5*).

Table 3: change "Ratio" to "Percentage" for consistence.

**Response:** Done (*please see P26 Table 3*)**.**

Line 23, pp6: it should be "in the same period" and "resulting the small net change"

**Response:** Done (*please see P10 L9*).

Line 28, pp8: change "classes" to "types"

**Response:** Done (*please see P14 L6*).

Line 11, pp9: what the difference between surface runoff and streamflow?

**Response:** In SWAT, surface runoff is flow that occurs along the sloping surface (i.e., quick-response overland flow), which contributed to streamflow. It occurs whenever the net rainfall exceeds the rate of infiltration or the it was saturated. Streamflow is the water flowing downhill through creeks, streams, and rivers toward the outlet of the basin, and it includes surface runoff, lateral flow within soil profile, and base flow.

Line 23, pp9: why cropland has the highest soil water storage?

**Response:** The reason can be explained as follows: Firstly, in the Yanhe basin, the cropland was mainly distributed in the bottom region of the valley and on land with gentle slope. The valley bottom generally receives more water flow from the hillslope and has lower temperature than higher regions, resulting in more soil moisture (Wang et al., 2012). Furthermore, land with gentle slope has higher soil water retention ability than does land with steep slope (Famiglietti et al., 1998). Secondly, our results showed that the cropland had lower evapotranspiration per unit area than did the other land use types. Thus, these interrelated above two reasons led to the highest soil water storage. This explanation has been added to the revised manuscript (*please see P14 L9 ~ P14 L15*).

Line 4, pp10: sloped cropland or sloping cropland? Pls make this term consistent throughout the manuscript.

**Response:** Thanks for the suggestion, and we used "sloping cropland" in the whole manuscript (*please see P16 L7-8*).

Line 13, pp10: "transition analysis"? Pretty new and never heard. Pls consider changing it. Otherwise more definition should be given.

**Response:** "Land us transition analysis" summarizes the amount of certain land that changes from each category at the initial time to each category at the subsequent time, and it has been widely used in landscape ecology and GIS studies of land use to quantitatively estimate the rate of change. To avoid confusion, we took "land use change analysis" in the revision (*please see P9 L20 & P26 L1*).

Line 24, pp10: change planting wood to afforestation?

**Response:** Done (*please see P17 L3*).

The study (Qiao et al., 2017, Woody plant encroachment reduce annual runoff and shifts runoff mechanisms in the tallgrass prairie, USA. Water Resource Research) provides some evidence of water budget difference between woodland and grassland and would be a useful reference for this study.

**Response:** Thanks for recommending a valuable reference to our study, and we have cited this article in our revision (*please see P14 L21-24*).

General Comment: This manuscript investigated the response of main hydrological components to land use change and basin management policy in the loess region of China. It is interesting to see the spatial distribution of hydrological responses at the unit (HRU) level and the quantitative assessment of the roles of different land use types in basin hydrology. In particular, it is attractive to interpret the mechanism of the streamflow decrease associated with vegetation restoration on the Loess Plateau and thus may have a broad readership. Overall, the article was well organized, well written, and easy to follow. All figures and tables are of high quality and informative. What I would suggest is give more discussion on the reason why the different land use types show different hydrological characteristics (e.g., soil water and ET) in this region. That would help reader better understand the findings from the article.

**Response:** We thank the reviewer for his/her positive remarks and constructive comments. We considered the comments of reviewer and added more discussion on hydrological characteristics. For soil water, explanations for differences among different land use types need to combine the topography and land cover on the land surface. In our study, woodland had lowest surface runoff and water yield, because woodland areas capture more rainfall and uptake more water than other land use types (e.g., cropland and grassland) (Wang et al., 2012; Jian et al., 2015), resulting in a lower infiltration, runoff, and water discharge (Wang et al., 2013; Duan et al., 2016). Qiao et al. (2017) reported that the reduced runoff in areas with woody plant relative to grassland areas is associated with shift in runoff generation mechanisms; thus, the shift from saturation excess overland flow to infiltration excess overland flow might also have contributed to the reduced runoff in woodland areas observed in the present study. Additionally, compared with other vegetation types, woodland areas might lose more water through ET. This pattern was evident from the change in soil water, with less soil water in woodland than in other areas under the same precipitation amount. Less soil water was observed in woodland because forests in the Yanhe basin generally grow on landform with high slopes; our analysis indicated that more than 62% of woodland was located on slopes ≥15° (Table 5). Steeper slopes generally retain less soil water due to low infiltration and rapid surface drainage (Famiglietti et al., 1998). Thus, a large amount of precipitation was associated with forest growth and ET rather than discharge out of the basin as surface runoff and streamflow. (*Please see P14 L16 ~ P15 L7*). Cropland had the highest soil water which can be partly attributed to the topography of cropland. The cropland area was mainly distributed in the bottom region of the valley and on land with gentle slope. The valley bottom generally receives more water flow from the hillslopes and has lower temperature than higher regions, resulting in more soil water (Wang et al., 2012). Furthermore, land with gentle slope has higher soil water retention ability than does land with steep slope (Panagopoulos et al., 2011). In addition, cropland had lower evapotranspiration per unit area than did the other land use types. In contrast, grassland had lower soil water, which can be attributed to its higher soil water loss through evapotranspiration. Wang et al. (2012) reported that grass cover types cannot protect the soil surface from solar radiation, leading to greater water loss via direct evaporation (*please see P14 L9 ~ P14 L16*). As for ET on the Loess Plateau region, it has been widely demonstrated that the woodland had highest ET, followed by grassland, and cropland (Xiao et al., 2013; Wang et al., 2012).

Specific comments:

P2 L6: Use "sloping cropland" here to guarantee the consistence, because you use "sloping land" in the whole manuscript.

**Response:** Corrected (*please see P16 L7-8*).

P7 L26 Add "of" between "trend" and "surface runoff".
**Response:** We rehearsed this sentence, and revised version is 'the decrease in surface runoff' (*please see P11 L22*).

P7 L29 Replace "because of the decreased area of cropland" by "because of its decreased area".
**Response:** Corrected (*please see P12 L2*).

P8 L1 Replace "deceasing" by "decrease".
**Response:** Corrected (*please see P12 L5*).

P9 L1-12 the response of soil water in woodland are well discussed, how about in cropland and grassland and try to explain more.
**Response:** As responded to the reviewer #1, highest soil water in cropland may be due to two related reasons: Firstly, in the Yanhe basin, the cropland area was mainly distributed in the bottom region of the valley and on the land with gentle slope, and valley bottom generally receives more water flow from the hillslope and has lower temperature than higher regions, resulting in more soil moisture (Wang et al., 2012); gentle sloping area had higher soil water retention ability compared with steep sloping area (Panagopoulos et al., 2011). Secondly, our results showed that the cropland had lower evapotranspiration per unit area than did the other land use types. Thus, these interrelated two main reasons led to the highest soil water storage in cropland. In contrast, grassland had lower soil water, which can be attributed to its higher soil water loss through evapotranspiration. Wang et al. (2012) reported that grass cover types cannot protect the soil surface from solar radiation, leading to greater water loss via direct evaporation (*please see P14 L9 ~ P14 L16*).

P9 L14 Rephrase the sentence and clearly state what are the main forms of land use change in the Yanhe basin.
**Response:** It was rephrased, and the revised sentence is "After the implementation of the GFGP, the area of cropland decreased continuously because it was transformed into grassland and woodland from 1990 to 2010, and conversions among cropland, grassland and woodland were the main forms of land use change in the Yanhe basin" (*please see P15 L8-10*).

P9 L20 Where are the locations of 26 watersheds?
**Response:** This study reported in the literature was conducted globally. The researchers selected 26 catchments around the world and evaluated the effect of afforestation on water yield. To avoid confusion, we added "globally" in this sentence (*please see P15 L15-16*).

P18: "Ratio" should be changed to "Percentage" in Table 3.
**Response:** Corrected (*please see P26 Table3*).

[revised manuscript text omitted]

---

## Author Response (AR2)

**Responses to editor and reviewers**

Dear editor and reviewers,

We greatly appreciate editor for providing opportunity again to revise our manuscript. We also thank two reviewers very much for pointing out some errors in writing. All comments are helpful for improving the quality of the manuscript. All authors have read and approved the revised manuscript. Please see our point-by-point responses below. A kindly note: reviewers' comments are in blue color and our responses are in black color; page and line numbers cited in our responses refer to the revised manuscript with track changes for easy evaluation. A track-changed version of the revised manuscript can be found after the response to reviewers.

**Reviewer #1:**

The authors have addressed the problems raised in my previous review. I would suggest an acceptance for publication after some minor revision. Below are some advices for consideration in their revision.

"Meteorological Institute of Shaanxi Province, Xi'an, China" Shanxi or Shaanxi?

**Response:** Shaanxi is correct. There are two provinces with same pronounce in China (Shǎnxi and Shānxi). To make the difference clear without tonal marks, the spelling "Shaanxi" was used to represent the Shǎnxi, while "Shanxi" is used for the province of Shānxi.

"Land use planning in China is considered crucial strategy for the sustainable management of a river basin systems and has.." change 'a river basin systems' to 'river basin systems'

**Response:** Done (*Please see P2 L13*).

"A geographic information system (GIS) interface,…" change it to "A Geographic Information Systems (GIS)"?

**Response:** Done (*Please see P6 L5*).

Pp11-12: the font sizes are different on these two pages.

**Response:** It has been corrected.

"Qiao et al. (2017) reported that the reduced runoff in areas with woody plant relative to grassland areas is associated with shift in runoff generation mechanisms; thus, the shift from saturation excess overland flow to infiltration excess overland flow might also have contributed to the reduced runoff in woodland areas observed in the present study." Change it to "Qiao et al. (2017) reported that the reduced runoff in areas with woody plant relative to grassland areas is associated with a shift in runoff generation mechanisms from saturation excess overland flow to infiltration excess overland flow. Such type of shift might also have contributed to the reduced runoff in woodland areas observed in the present study."

**Response:** It has been corrected accordingly (*Please see P12 L23-25*).

"Furthermore, afforestation on sloping land had greater hydrological effects than did planting grass" change this sentence to "… afforestation had greater hydrological effects than grass planting on sloping land"?

**Response:** It has been corrected accordingly (*Please see P15 L1*).

"Table 5 Slope characteristics of each land use type in 2010 and under different slope land conversion scenarios (unit: km2 )" Since they are scenarios, how can they be in the certain year? More revision to the caption?
**Response:** Following the reviewer's suggestion, the caption was revised, and new version is "Slope characteristics of each land use type at baseline and under different slope land conversion scenarios" (unit: km$^2$) (*Please see P26 L1*).

**Reviewer #2:**
I think the revised manuscript has adequately addressed all my comments and suggestions. However, minor revisions are needed before acceptance for publications
P 2, Line 4: "soil moisture" or "soil water"? Please keep consistent throughout the MS.
**Response:** We have replaced "soil moisture" by "soil water" to keep consistent (*please see P2 L5; P13 L25; P14 L9*).

P 2, Line 7-8: What is the difference between "land use type" and "LULC". I don't find the difference between the "variation in land use type" and "spatial heterogeneity of LULC". Please clarify.
**Response:** Thanks for pointing out the confused usage. In the manuscript, the "land use type" and "LULC" actually express the same meaning, thus the sentence "the characteristics of basin hydrology vary among land use patterns due not only to the type of LULC but also to the spatial heterogeneity of LULC" was corrected to "the characteristics of basin hydrology vary among land use patterns due not only to the type of LULC but also to the spatiotemporal heterogeneity of LULC" (*Please see P2 L8-9*).

P 4, Line 6-7: Delete this sentence.
**Response:** Done (*Please see P4 L9*).

P 4, Line 15: is characterized by
**Response:** Done (*Please see P4 L18*).

P 4, Line 18: Change "from 1952–2015" to "during 1952–2015"
**Response:** Done (*Please see P4 L21*).

P 4, Line 22: "from 1952 through 2008"?
**Response:** Yes, the hydrological data was available from 1952 to 2008, and we replaced "through" by "to" (*Please see P5 L1*).

P 6, Line 18: Change "In order to" to "to"
**Response:** Done (*Please see P6 L22*).

P 8, Line 2: Change "increased slowly" to "slowly increased"
**Response:** Done (*Please see P8 L6*).

P 8, Line 24: Change "highest" to "largest".
**Response:** Done (*Please see P9 L5*).

P 9, Line 6: Change "shows" to "showed".

**Response:** Done (*Please see P9 L12*).

P 9, Line 9: Delete "highest".

**Response:** Done (*Please see P9 L14*).

P 9, Line 16: Change "presents" to "presented".

**Response:** Done (*Please see P9 L21*).

P 9, Line 22: "1990-2000" or "1990–2000"? Please check the dash line.

**Response:** Thanks, we checked the usage of hyphen "-" and dash "–" carefully, and all errors were corrected in the manuscript (*Please see P10 L3; P8 L9-11; P13 L16*).

P 9, Line 23: Change "are" to "were"

**Response:** Done (*Please see P10 L5*).

P 11, Line 5-7: Change "illustrates" to "illustrated"; "decreases" to "decreased"; "are" to "were"

**Response:** Done (*Please see P11 L11*).

P 12, Line 11-12: Change "attributed to the topography of cropland" to "attributed to its topography"

**Response:** Done (*Please see P12 L12*).

Reference: The format should keep consistent with HESS.

**Response:** The format of reference was controlled by EndNote software, and we downloaded the output style from HESS to keep all bibliography consistent with journal requirement.

Table 2 and 3: Please explain the "Pct". Alternatively, you can write the full name of "percentage".

**Response:** we added the note below the table 2 to explain the meaning of the "Pct."(*Please see P23 Table 2*).

Fig. 1: Add the unit of the legend of the DEM

**Response:** The unit was added in the Fig.1 (*Please see P28 Fig. 1*).

[revised manuscript text omitted]